



# Assessing the rotor blade deformation and tower-blade tip clearance of a 3.4 MW wind turbine with terrestrial laser scanning

Paula Helming[1], Alex Intemann[1], Klaus-Peter Webersinke[2], Axel von Freyberg[1], Michael Sorg[1], Andreas Fischer[1,3]

[1]University of Bremen, Bremen Institute for Metrology, Automation and Quality Science, 28359 Bremen, Germany
[2]LASE Industrielle Lasertechnik GmbH, 46485 Wesel, Germany
[3]BEST Bremen Research Centre for Energy Systems, University of Bremen, 28334 Bremen, Germany

*Correspondence to*: Paula Helming (p.helming@bimaq.de)

**Abstract.** Wind turbines have grown in size in recent years, making an efficient structural health monitoring of all of their

structures ever more important. Wind turbine blades deform elastically under the loads applied to them by wind and inertial forces acting on the rotating rotor blades. In order to properly analyze these deformations, an earthbound system is desirable that can measure the blade deformation, as well as the tower-blade tip clearance from a large measurement working distance of over 150 m and a single location. To achieve this, a terrestrial laser scanner (TLS) in line-scanning mode with vertical alignment is used to measure the distance to passing blades and the tower for different wind loads over time. In detail, the

blade deformations for two different wind load categories are evaluated and compared. Additionally, the tower-blade tip clearance is calculated and analyzed with regards to the rotor speed. Using a Monte-Carlo simulation, the measurement uncertainty is determined to be in the mm-range for both the blade deformation analysis and the tower-blade tip clearance. The in-process applicable measurement methods are applied and validated on a 3.4 MW wind turbine with a hub height of 128 m. As a result, the deformation of the blade increases with higher wind speed in wind direction, while the tower-blade tip clearance

decreases with higher wind speed. Both relations are measured not only qualitatively but also quantitatively. Furthermore, no difference between the three rotor blades is observed, i.e. each of the three blades is shown to be separately measurable. The tower-blade tip clearance is compared to a reference video measurement, which recorded the tower-blade tip clearance from the side, with validated the novel measurement approach. Therefore, the proposed setup and methods are proven to be effective tools for the in-process structural health monitoring of wind turbine blades.

## 1. Introduction

Wind turbines have increased in height and diameter over time leading to increasing loads that need to be tolerated during their lifetime. The resulting dynamic behavior affects the turbine's stability, service life and efficiency. In particular, the rotor blades experience a complex load as a result of wind forces and rotor dynamics. In addition, due to their increasingly thin shape, they grow more vulnerable to deformations and material fatigue. While simulation models can predict the load situation of the rotor

blades for an assumed boundary condition, it is only an approximated and fragmentary impression of the real load on-site





during the actual turbine's operation. Additionally, the simulation models in question are typically not validated by field measurements of rotor blade deformations. For validating load simulation models as well as evaluating and predicting the structural condition of tall wind turbines during operation, on-site and in-process measurements of dynamic rotor blade deformations are indispensible.

The measurement methods commonly used for this purpose often need a wind turbine modification, which leads to high efforts and costs since they have to be installed during standstill. Additionally, sensors installed on the rotor blades, such as strain gauges (Crabtree et al. 2014) or accelerometers (Loh et al., 2017; Oliveira et al., 2018; Ou et al., 2017; Tcherniak and Mølgaard, 2017), can lead to altered structural behavior which can impair the validity of the measurement results. Also, these sensors measure the deformation only at a few discrete locations, and changing these points of installation means a turbine modification

once more.

For this reason, there have been several attempts to contactlessly measure the rotor blade deformation. One is the use of laser Doppler vibrometry, where the frequency and phase change of a laser beam pointed at the rotor blade is measured to determine the rotor blade oscillations (M Ozbek et al., 2009, Dilek et al., 2019). However, the installation of retroflective markers at discrete measuring spots is needed. One method that could be used without the need of external markers is photogrammetry.

Here, the three-dimensional coordinates of the rotor blades are determined from multiple two-dimensional images taken from different positions and orientations by using either point-tracking on wind turbine models (Lundstrom et al., 2012), point tracking on actual operating wind turbines (Ozbek et al., 2010; Ozbek and Rixen, 2013; Ozbek et al., 2013) or using digitial image correlation (DIC) (Winstroth et al., 2014; Lehnhoff et al., 2020). However, several different measurement locations or access points are needed to measure the rotor blade geometry, as well markers on the rotor blade.

In contrast, terrestrial laser scanners (TLS) offer the potential to contactlessly measure the wind turbine deformation from a single access point, from distances > 150 m and with no blade modifications. While TLS have been proven to be able to measure the tower's deformation by aligning the laser scanner either vertically (Schill and Eichhorn, 2016; Artese and Nico,2020) or horizontally (Helming et al., 2021) to the tower, only few attempts to measure rotor blade characteristics took place. Goering et al. (Grosse-Schwiep et al., 2014; M. Goering and T. Luhmann, 2020) used point-based TLS combined with

an optical camera placed on the nacelle to measure the rotor blade torsion and deflection during a rotation by comparing it to a geometric model of the blade. Hoghooghi et al. 2020 used a rotational platform to follow the blade and measure the displacement of one point on the rotor blade at a time during the rotation by using the information gained from the Supervisory Control and Data Acquisition (SCADA) system. However, both Goering et al. and Hoghooghi et al. methods, require prior knowledge of wind turbine geometry or SCADA-data as well as additional equipment such as a camera or rotational platform

besides the laser scanner. And, terrestrial laser scanners have not yet been applied to assess the rotor blade deformation and the tower-blade tip clearance for different loads.

Therefore, the aim of the present article is a TLS-based correlation analysis between the applied wind loads on the wind turbine and the rotor blade deformation as well as the tower-blade tip clearance during operation, using neither any prior knowledge of the geometry of the wind turbine nor access to the SCADA data. The correlation analysis is realized by a statistical analysis





of the TLS data, which is explained together with the TLS measurement arrangement and principle in section 2. The experimental setup is described in section 3, and the achievable measurement uncertainty is estimated in section 4 by a Monte Carlo simulation. The measurement system is then applied to a 3.4 MW wind turbine, where the deformation of the rotor blade is analyzed for different loads and the measured tower-blade clearance is compared to a reference measurement system in section 5, section 6 closes with a summary.

## 2. Measurement principle and methodology


The TLS used for the present analysis is a scanner in lane scanning mode. The measurement principle and the measuring approach are presented in the following sub-sections.

### 2.1. Measurement principle

Terrestrial laser scanning (TLS) measures a distance $d$ with the time-of-flight (ToF) principle: A pulsed laser beam is emitted,
and the time $t$ it takes for the light to reach the target and return to the laser scanner is measured. With the speed of light $c$, the distance $d$ is then obtained by calculating

$$d = \frac{c \cdot t}{2}. \tag{1}$$

This distance measurement is repeated while the direction of the emitted laser beam direction is varied, i.e. scanned. Terrestrial laser scanners in line-scanning mode have a rotating mirror that scans the laser beam over a fixed scan angle $\theta$. Each scan $S_i$, where $i$ is the scan number, is related to a time $\tau_i$ and consists of $N$ measured distances $d_i$, which are acquired
in incremental steps with a constant angular step width $\Delta\theta = \frac{\theta}{N}$:

$$S_i = \begin{pmatrix} d_i(\Delta\theta) \\ \vdots \\ d_i(N\,\Delta\theta) \end{pmatrix}.$$

As a result, the surface points are first obtained in polar coordinates and, thus, are transformed to two-dimensional Cartesian coordinates (depth axis and scan axis).

### 2.2. Methodology

The laser scanner in line scanning mode is aligned vertically, scanning the tower and blades of the wind turbine as they pass through the scan line as displayed in Figure 1.





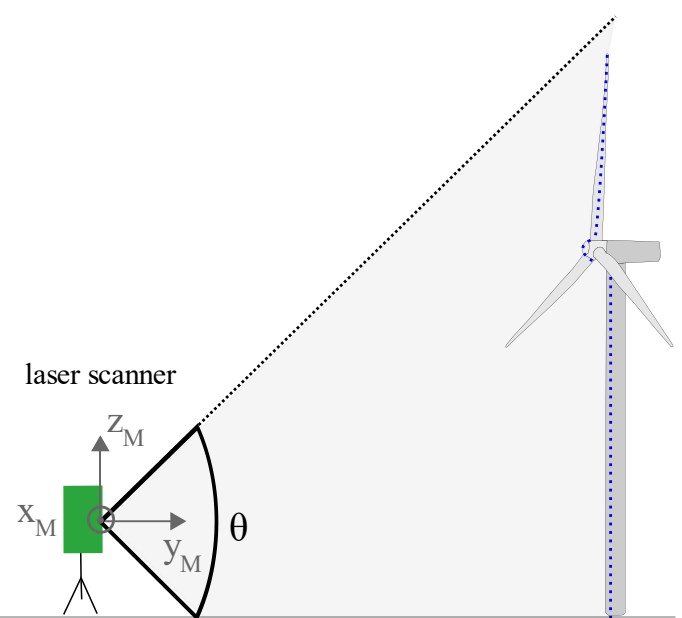

**Figure 1: Sketch of the measurement arrangement: the laser scanner (in green) with the scan angle $\theta$ is aligned vertically to the**
**wind turbine. The points that are reflected by the wind turbine (blue dots) are measured in the $y_M$, $z_M$ coordinate system.**

### 2.2.1.    Data segmentation

The wind turbine surface points are measured continuously over a certain measurement period during operation, particularly

over multiple rotor revolutions. A rough estimate of the distance between the laser scanner and the wind turbine is used for a

threshold to remove points far from the wind turbine, which is performed for each scan $S_i$ of the raw measurement data. Each

rotor blade is scanned several times as it passes through the scan line for each rotation, however, only the longest blade scan

vector, i. e. the scan $S_i$ with the highest number of points on the blade for one rotation and thus the most information of the

blade, is considered further. The remaining data points are then partitioned into: tower (blue), upper blade (orange), lower

blade (yellow) and nacelle (grey) by using their location in the measurement coordinate system, see Figure 2. The blade scans

are further subcategorized into blade 1, blade 2 and blade 3 of the wind turbine by identifying those scans with no blade points

present, which separate the consecutive blades.

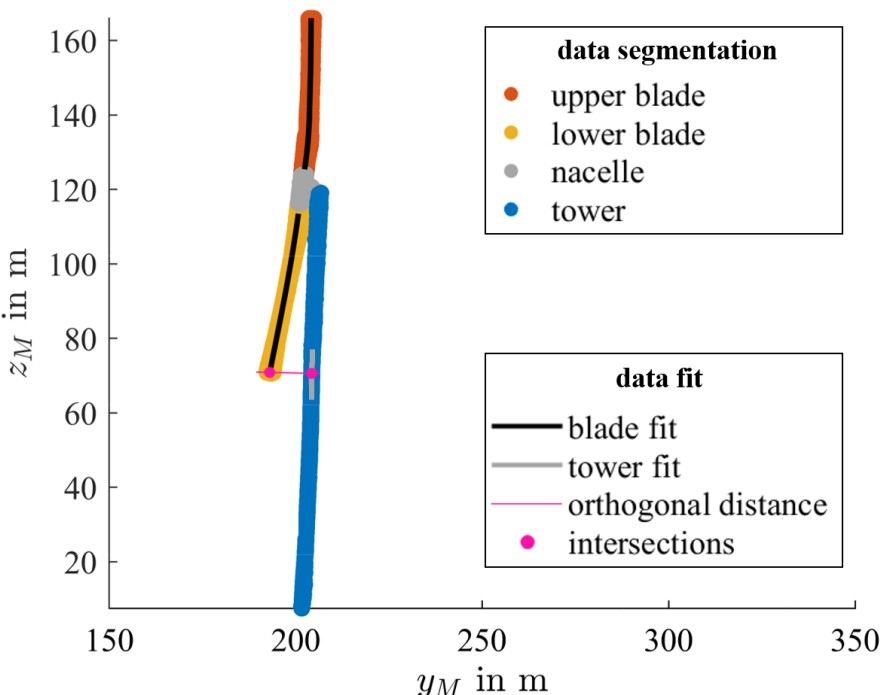

**Figure 2: Results of data segmentation: The assignment of the valid measuring data is color-coded according to the legend. Furthermore, the fitting results of the tower and the rotor blade data points are presented, together with the intersections used to calculate the orthogonal distance between the lower blade tip and the tower.**

### 2.2.2. Rotor blade deformation

Since no additional equipment such as a co-rotating platform or optical camera is used in the measurement approach shown in Figure 1, it is not possible to continuously measure the rotor blade deformation during the rotation. Therefore, a statistical analysis is carried out to analyze the correlation between different wind loads and the rotor blade deformation at the position of the scan line. With the TLS-system, it is not possible to measure the wind speed directly. However, since the rotor speed $\omega$ increases linearly with the wind speed in partial load operation mode of the turbine, the rotor speed can serve as an indirect measure of the wind load. To do this, the rotor speed $\omega$ is calculated from the time difference between two consecutive rotor blades passing the scan line. Furthermore, all blade scans are then divided into two categories: a) higher and b) lower or equal than the mean rotational speed $\omega_{\mathrm{mean}}$ during the measurement period. As shown in Figure 2 as a black line, a fourth-degree polynomial is then fitted using the least-squared-method through the points of all blade scans of blade 1 in each rotational speed category (blade-fit), separately for the upper and lower blade position. The resulting fit represents the mean deformed rotor blade 1 for each category, so that an analysis of the blade deformation with regards to different wind loads is possible. The described evaluation is repeated for blade 2 and blade 3, i. e. for each blade of the wind turbine.



### 2.2.3. Tower-blade tip clearance

To measure the tower-blade tip clearance, defined as the orthogonal distance between the tower and the tip of the lower rotor
blade, a fourth-degree polynomial is fitted through the points on each lower blade (blade-fit) similar to the method for the rotor
blade deformation. In contrast to the evaluation of the rotor blade deformation, however, the fit is performed for each individual
lower blade scan. Then, a point of interest near the blade tip is defined and, within the shortest time interval to the blade scan,
the scan number $i$ is selected, which contains at least five points on the tower around the projected point of interest on the
tower. Through these tower points, a straight line is fitted around the $z_M$-coordinate of the point of interest, see grey line in
Figure 2 (tower-fit). Finally, a line between the tower-fit and the blade-fit is constructed, which is orthogonal to the tower-fit.
The distance (pink line) between the resulting intersections (pink dots) with the tower and the blade, respectively, is defined
as the tower-blade tip clearance, see Figure 2. Note that the local thickness of the blade has to be subtracted from this distance
to obtain the actual tower-blade tip clearance.

## 3. Experimental setup

### 130 3.1. Measurement object

Measurements are carried out on a 3.4 MW wind turbine of the type REPower 3.4M located in Bremen, Germany. Some of
the wind turbine characteristics are summarized in Table 1.

**Table 1: Wind turbine characteristics**

| hub height | 128 m |
|---|---|
| rotor diameter | 104 m |
| cut-in wind speed | 3.5 m/s |
| rated wind speed | 13.5 m/s |
| cut-out wind speed | 25 m/s |

### 135 3.2. Measurement system and setup

#### 3.2.1. Setup

A photo of the experimental setup of the laser scanner aligned vertically to the wind turbine and connected to a measurement
laptop is shown in Figure 3 left. The laser is placed upwind from the wind turbine facing it from the front with a distance of
190 m in the first and 200 m in the second experiment. This setup is used for measuring both the rotor blade deformation as
well as the tower-blade tip clearance. For validation, the tower-blade tip clearance is simultaneously measured with a camera
system as a reference using the method and setup described in He et al., 2013. In contrast to the laser scanner, the video camera





is placed at a 90° angle, looking at the wind turbine from the side, see Figure 3, right. The tower-blade tip clearance is then determined by measuring the pixels between the blade and the tower whenever a blade reaches the lowest point, this distance in pixels is then transformed to meters using the known length of the nacelle as a calibration factor. Note that no video
measurement for the rotor blade deformation experiment was performed.

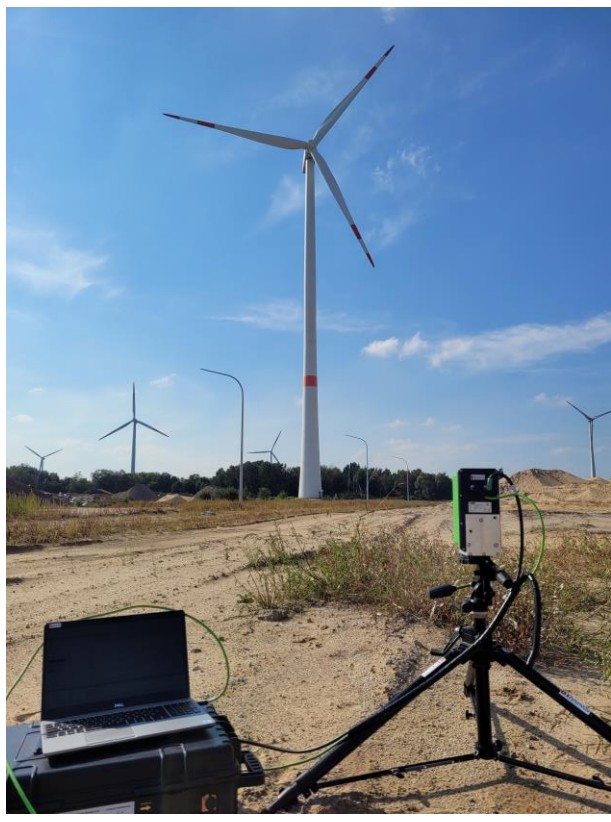
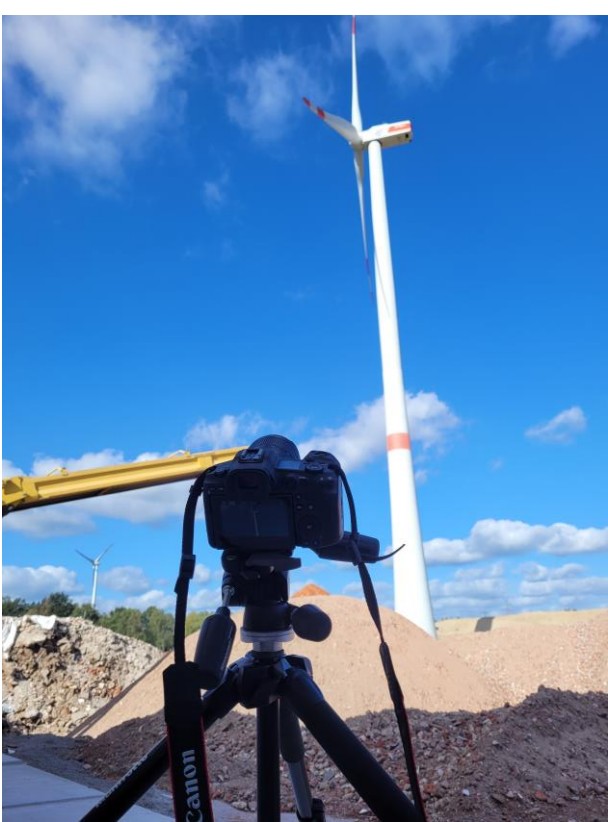

**Figure 3: Left: experimental setup for field measurements with a laser scanner on a tripod, a measurement laptop and the wind turbine in the background. Right: experimental setup for field measurement with a camera on a tripod and the wind turbine from the side in the background used as a reference for the tower-blade tip clearance.**

**3.2.2.    Measurement system**

The laser scanner used in this study is the LASE 200D-228PLUS using a laser in the infrared range. It produces 1000 measurement points over a scan angle of $\theta = 90°$, which leads to an angular step width of $\Delta\theta = 0.09°$. The scan frequency is 30 Hz. The relevant characteristics of the laser scanner are summarized in Table 2.





**Table 2: Laser scanner characteristcs**

| laser wave length | $\lambda = 905\ \text{nm}$ |
|---|---|
| laser pulse rate | 40 kHz |
| laser class | 1M |
| laser spot size at sensor window | 12 mm x 16 mm |
| scan angle | $\theta = 90°$ |
| points per scan | $N = 1000$ |
| scan rate | 30 Hz |

The optical camera used for the reference measurements of the tower-blade tip clearance is a Canon EOS R5 system camera, the relevant camera characteristics are summarized in Table 3.

**Table 3: Camera characteristcs**

| frame rate | 25 Hz |
|---|---|
| effective pixels | 40 Megapixel |
| maximum resolution | 8192 pixels x 5464 pixels |

### 3.2.3. Data processing

The data of each scan is saved as a text file and imported to MATLAB, where the data processing is carried out. For the calculation of the rotor speed, all scans $S_i$ with measurement points on the blades are considered, even the scans with incomplete blade scan vectors. For each $\Delta\theta$ where every blade is expected to pass through the scan line, the first point of contact for each blade with the scan line is determined and the time $t_i$ of the respective scan $S_i$ is recorded. Afterward, the

rotor speed is calculated from the time difference $\Delta t$ between consecutive blades passing through the scan line for each $\Delta\theta$. For each 10 s interval the calculated rotor speed is averaged, yielding rotor blade speed data with a time resolution of 10 s.

The polynomial fits for both the rotor blade deformation analysis and the tower-blade tip clearance are done as least-squares-fittings using the MATLAB plot fitting toolbox. To calculate the tower-blade tip clearance, the lowest $z_M$-coordinate which is present in all blades is used as the point of interest on the tower, and a region $\pm 3$ m in $z_M$ is used to fit a line through the tower.

A further line, which is orthogonal to tower fit curve and passes through the point of interest, is constructed intersecting the blade fit. The length between the point of interest and the intersection point is defined as the tower-blade tip clearance. Note that the local thickness should be subtracted from the tower-blade tip clearance, which is not applied here since the thickness is unknown and the focus of this study is the analysis of the clearance fluctuations. The data for the tower-blade tip clearance is smoothened using a moving median with a window size of 3 seconds.






### 3.3. Measurement conditions

The measurement conditions for the two measurement campaigns, where each was performed regarding one of the two measurement tasks (rotor blade deformation, tower-blade tip clearance), are summarized in Table 4. Both measurements took place for around 30 minutes. For the last 18 minutes of the tower-blade tip clearance measurement, the clearance was also
measured with the reference camera measurement system. The measurements were carried out during partial-load operation of the wind turbine to avoid influences of pitching of the rotor blades in the data. No yaw-movement occurred during the measurement period.

**Table 4: Measurement conditions for the different experiments**

|  | **rotor blade deformation** | **tower-blade tip clearance** |
|---|---|---|
| date | 2021-09-14 | 2022-03-04 |
| measurement length | 30:36 min | 30:00 min (last 18 min additionally with video system) |
| distance laser scanner to tower | 200 m | 190 m |
| distance video to tower | -- | ~ 170 m |
| rotor speed range | 7.4 rpm – 10.2 rpm | 8.4 rpm – 10.0 rpm |
| average wind speed | 3.9 m/s | 5.9 m/s |

## 4. Uncertainty analysis

### 4.1. Uncertainty of the laser scanner

To determine the uncertainty for one measurement point of the laser scanner, a white board with similar reflective characteristics as the rotor blade is placed at distances $s$ from 50 m to 300 m with a step size of 50 m, see Figure 4. The board is also scanned at 310 m, which is the largest specified measuring distance with the used laser scanner. Additionally, measurements with different tilt angles $\gamma$ of the board are performed, i. e. for $\gamma = 0°$, $\gamma = 15°$ and $\gamma = 30°$. For each
constellation, $i > 10{,}000$ scans are conducted, and the standard deviation of one point $d(\Delta\theta)$ is calculated. Note that the systematic error of the laser scanner is not considered, since not the absolute distance measurements but only the distance changes or a relative distance w.r.t. the measured distance at a different scan position is of interest. Thus, the standard deviation yields the random error of one point of the laser scanner, which is studied here in dependency of the working distance.



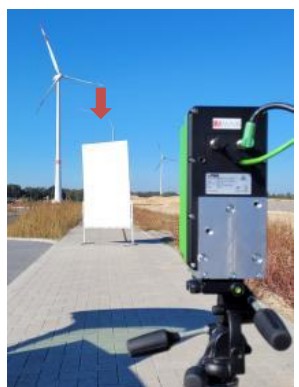
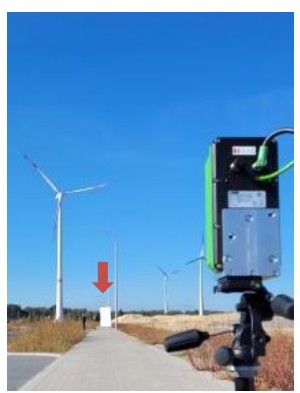
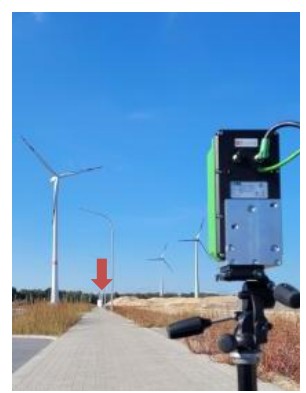

**Figure 4: White board (marked with red arrow) placed at different working distances and with different angles with respect to the laser scanner. The number of scans for each position and orientation amounts to > 10,000.**

Figure 5 shows the calculated standard deviation over the working distance $s$ for the tilt angle $\gamma = 0°$ in blue, $\gamma = 15°$ in red and $\gamma = 30°$ in yellow, respectively. A linear polynomial function is fitted through each dataset and is represented as a dotted line. The standard deviation increases linearly with the distances as described in (Freyberg et al., 2021) and increases with an

increased angle $\gamma$.

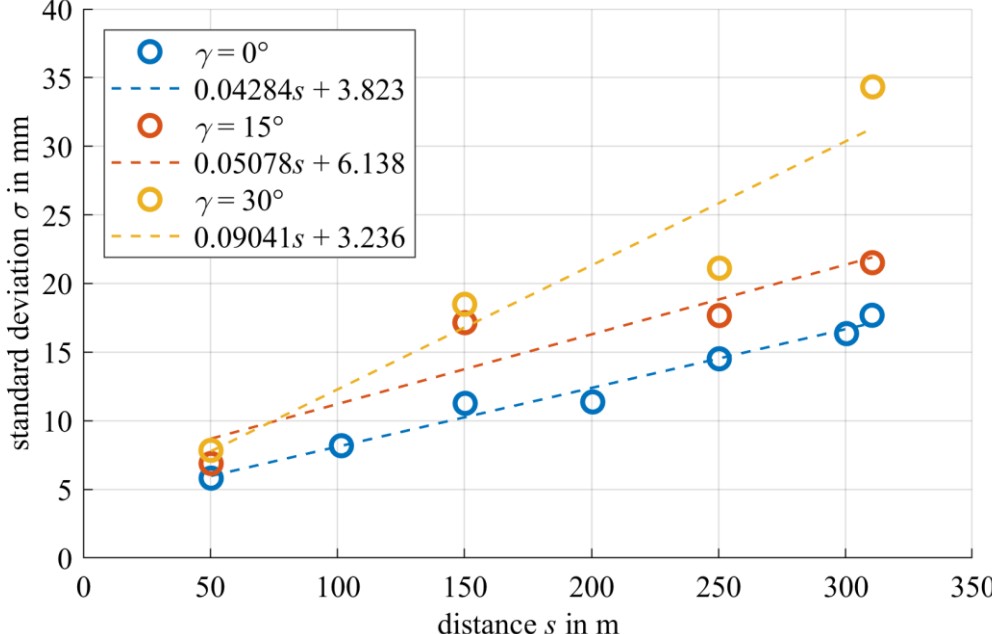

**Figure 5: Standard deviation of the measured distance to a white board over the working distance $s$ for different board tilt angles $\gamma$ (circles). A line is fitted through the points (dottet lines).**





## 4.2.    Uncertainty of rotor blade deformation fit

In order to determine the uncertainty of a rotor blade fit concerning the rotor blade deformation analysis, the determined behavior of the standard deviation of a single measurement point from Figure 5 is now used to perform a Monte Carlo simulation according to (Joint Committee for Guides in Metrology, 2008). Due to the twists in the rotor blade, it is hardly possible to predict the exact angle between the laser beam and the rotor blade surface. Therefore, three different Monte Carlo simulations are carried out, each assuming that the rotor blade is hit by the laser beam at a constant angle, i. e. $\gamma = 0°$, $\gamma = 15°$ or $\gamma = 30°$. As the basis for one run of the Monte Carlo simulation, 92 measuring points are simulated to represent the upper blade and 83 measuring points representing the lower blade, which is equivalent to the number of points in the actual blade measurement that is contained in one scan. Normally distributed noise is added to each data point according to its nominal distance to the laser scanner. To complete one run of the Monte Carlo simulation, this generation of simulated measurement data is repeated 263 times, and an example of the resulting data set is shown in Figure 6 (blue circles). The amount of data is equivalent to the number of blades measured in the measurement period for the category of the higher rotational wind speeds $\omega > \omega_{mean}$. Note that the simulation does not take effects from the blades' movement into account. Through all simulated measuring points (92 x 263 for the upper blade and 83 x 263 for the lower blade), a fourth-degree polynomial function is fitted through the points, separately for the upper and the lower blade (black lines in Figure 6), as described in the method section 2.2.2.

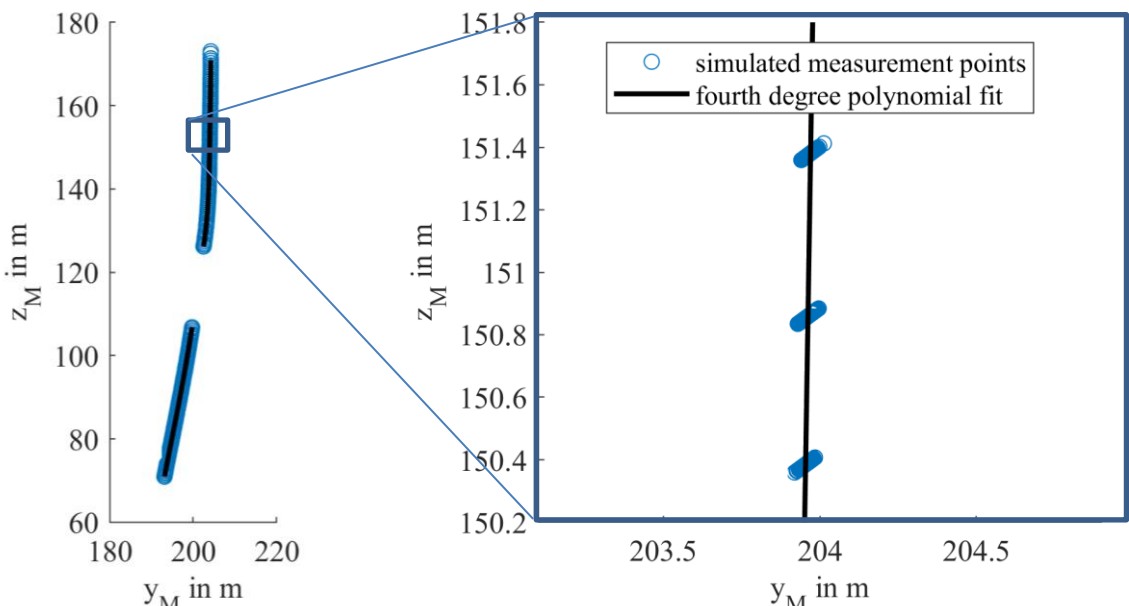

**Figure 6: Simulated points on blades for 263 scans with noise added to each simulated measuring point (blue circles) for the case $\gamma = 0°$. A fourth-degree polynomial is fitted through the simulated points (black line).**





One run of this simulation is repeated 10,000 times and the standard deviation of the fit in $y_M$-direction as a function of $z_M$ is calculated and shown in Figure 7. The standard deviation increases with an increasing tilt angle $\gamma$. The standard deviation is fairly constant in the middle of the blades but increases rapidly at the end points of the blades. Due to the divergence of high-order polynomial functions outside of the interval considered for interpolation over a set of equispaced points (see (Boyd and Xu, 2009)), the residue of a least-squares approximation in turn increases at the edges of the fit interval, leading to a rapidly increasing standard deviation for points approaching the edges of the blades as seen in Figure 7. If the start and end points of

the blades are ignored, the uncertainty of the fit resulting from the uncertainty of the laser scanner is below 0.3 mm, even in the scenario where the laser beam hits the rotor blade at an angle of $\gamma = 30°$, and even with the endpoints included it reaches a maximum of less than 0.7 mm. Using the laser scanner, it is therefore possible to differentiate between two deformed states in the mm range.

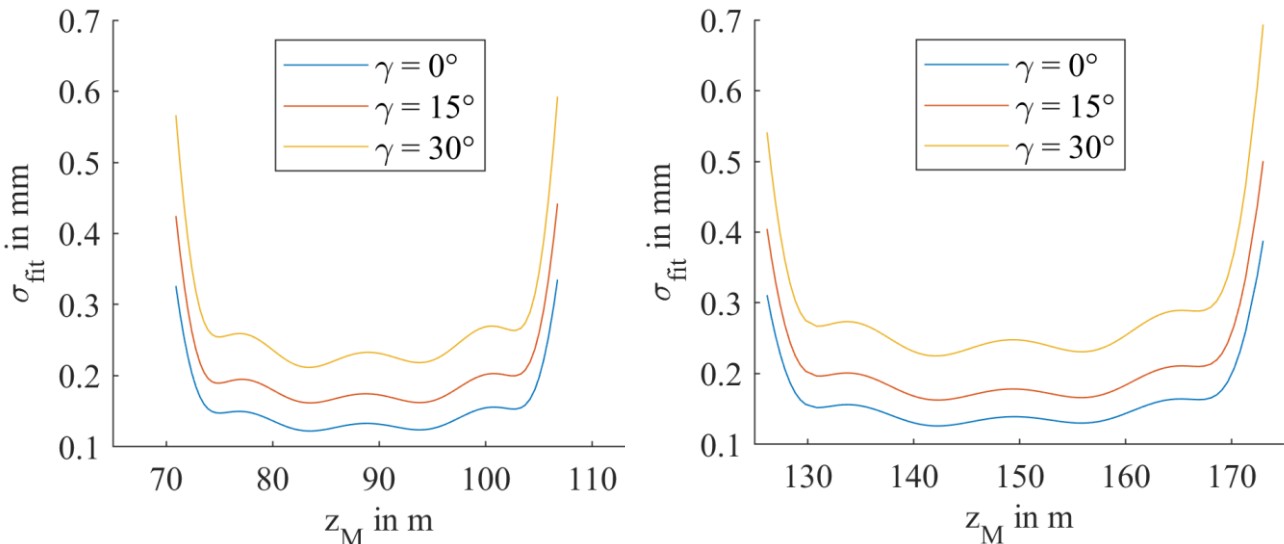

**Figure 7: Standard deviation of the fourth-degree polynomial fit along the $z_M$-axis for the lower blade (left) and upper blade (right)**
**through 263 simulated rotor blade scans repeated 10,000 times.**

### 4.3.    Uncertainty tower-blade tip clearance

To determine the uncertainty of the rotor blade fit for the tower-blade tip clearance, another three Monte Carlo simulations are carried out with different angles $\gamma$. Similar to section 4.2, rotor blades are simulated, but only the measurement points corresponding to the lower blade are simulated and added with noise, see Figure 8. Additionally, the measurement points for

one tower scan are also simulated and added with normally distributed noise. As described in 2.2.3, a fourth-degree polynomial is fitted through the simulated blade measurement, a line is fitted through the tower measurement and the line intersecting both fits is calculated to finally obtain a measure of the tower-blade tip clearance. This is repeated 10,000 times and the standard deviation of this calculated tower-blade tip clearance is computed and shown in Figure 9. The uncertainty is a magnitude higher compared to the uncertainty of the blade deformation fit and increases with increasing angle $\gamma$. This can be explained





by the fact that only the simulated measurement points of one blade and one tower scan are used for this calculation which increases the uncertainty. Additionally, the tower-blade tip clearance uses the tip of the rotor blade fit, which has the highest uncertainty as shown in section 4.2. Nevertheless, the tower blade-tip clearance can be calculated with an uncertainty of around 1 cm, even when the laser scanner hits the surface at an angle of $\gamma = 30°$.

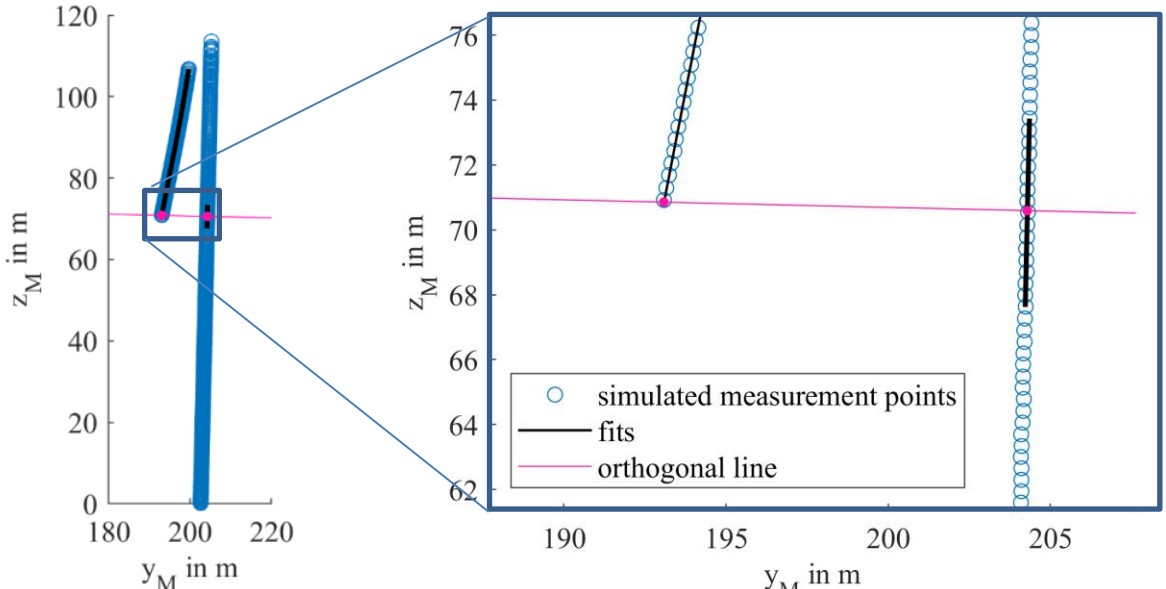

**Figure 8: Simulated measuring points on blade and tower for one scan with noise added to the blade and tower points (blue circles)**
**for the case $\gamma = 0°$ with fits through simulated points in black. The orthogonal distance with the tower and blade is shown in pink.**

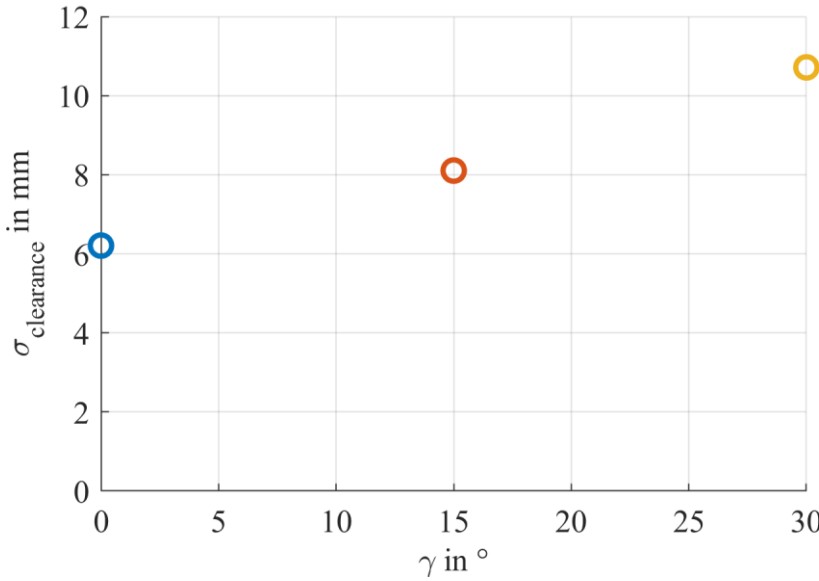

**Figure 9: Standard deviation of the simulated tower-blade tip clearance from 10,000 repetitions for different angles $\gamma$.**





## 5. Measurement results

### 5.1. Rotor blade deformation

To measure the deformed blade, a polynomial is fitted through the measurement points on the blade. Figure 10 shows all the measured points on blade 1 for all rotor speeds as colored circles. The fourth-degree polynomial fitted through all these points is shown in black, separately for the upper and lower blade. In the left graph, the axes have equal spacing, whereas in the right graph the view is distorted by purpose to make the different measurement points visible. In Figure 11, the polynomial fits for the measurement period is shown for the three different blades (blade 1 in blue, blade 2 in orange, blade 3 in green). Even in

the distorted view on the right, there is no visible difference between the blades, which leads to the conclusion that there is no defect or other characteristic on one of the blades causing it to behave differently on average than the other blades. Nonetheless, the following evaluation is carried out separately for the respective rotor blades in order to eliminate the influence of the different properties of the individual blades.

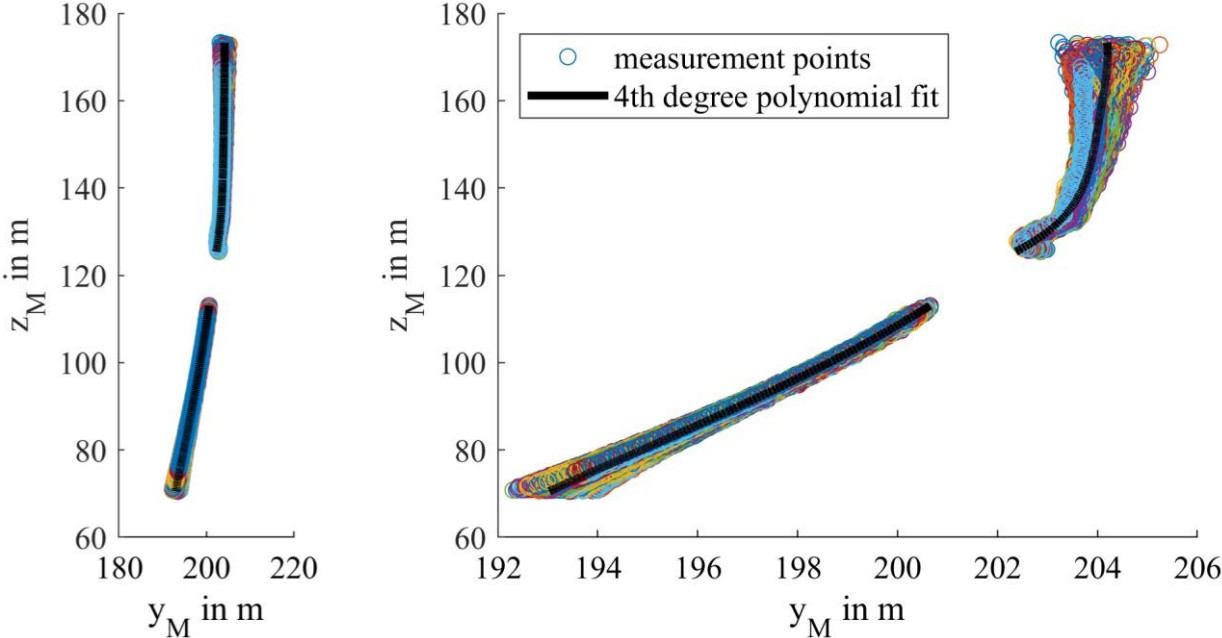

**Figure 10: Measurement points on the blades as circles and 4th degree polynomial fits for blade 1 for all rotor speeds.**

In order to analyze the rotor blade deformation for different wind loads, the scans are sorted according to the rotor speed at the time and a fourth-degree polynomial is fitted separately through all measured blade points for each category, as described in sections 2.2.2. and 3.2.3. The measured rotor speed $\omega$ is shown over the measurement time in Figure 12 with the mean rotor speed $\omega_{mean}$ shown as a dotted line. To show the difference between higher wind load and lower wind load, each scan is sorted either into a) higher (shaded in orange) and b) lower or equal (shaded in blue) than the mean rotational speed

$\omega_{mean} = 8.6$ rpm.



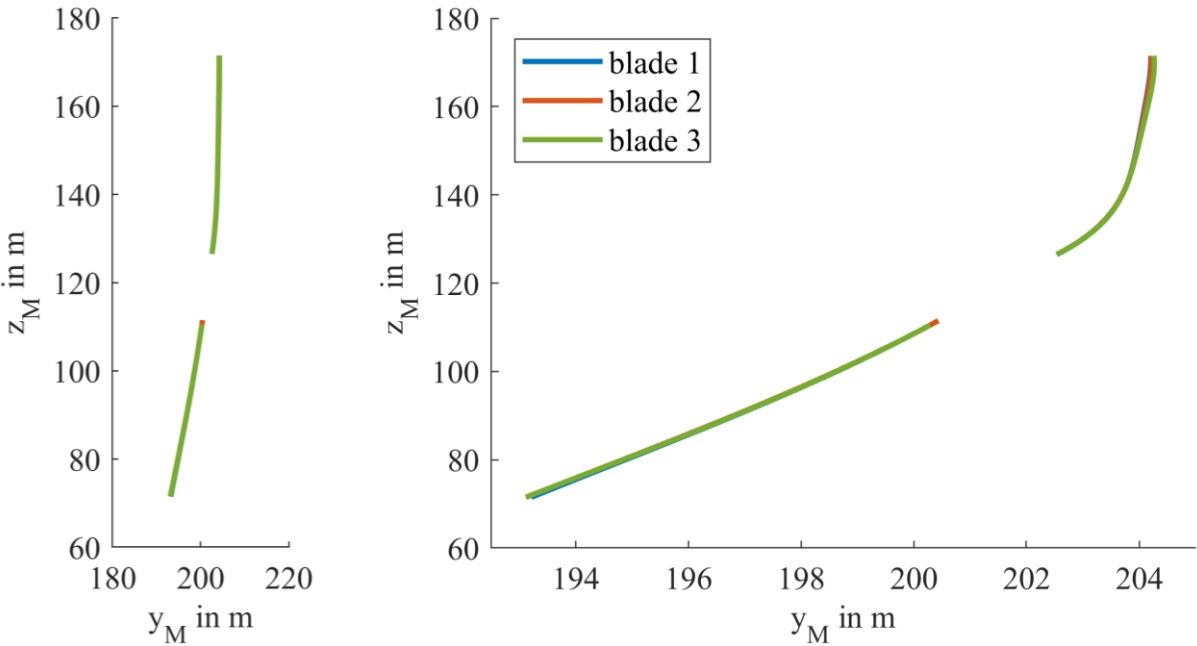

**Figure 11: Blade deformation for different blades: 4th-degree polynomial fitted through all measurement points of the separate blades with equal spacing between the $x_M$- and $z_M$-axis (left) and distorted view (right).**

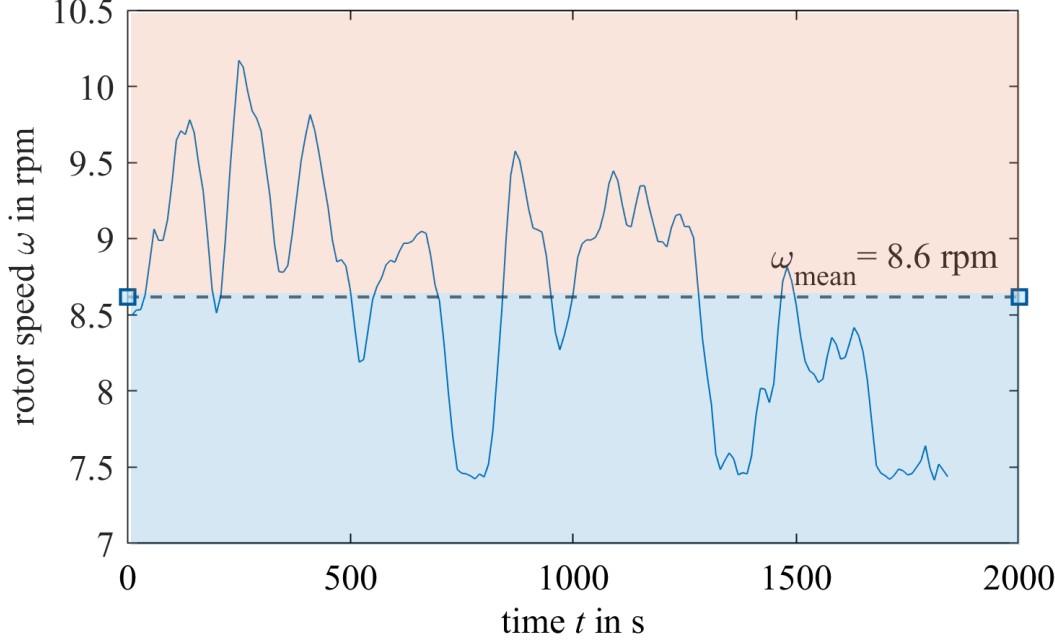

**Figure 12: Measured rotor speed $\omega$ over the measurement time with $\omega_{mean} = 8.6$ rpm. Each scanned rotor blade is categorized into higher wind loads ($\omega > \omega_{mean}$) shaded in orange or lower wind loads ($\omega \leq \omega_{mean}$) shaded in blue.**


Figure 13 shows the polynomial fit for lower wind loads ($\omega \leq \omega_{\text{mean}}$) in blue and higher wind loads ($\omega > \omega_{\text{mean}}$) in orange for blade 1. The other blades show similar results and are therefore not shown here. The distance in $y_M$-direction between the blades of the two wind force categories increases towards the tip of the blade, reaching around 0.4 m at the tip of the lower blade and 0.6 m at the tip of the upper blade. The light blue and light orange area show the standard deviation as a measure for

the amplitude of the oscillations of the blade. The amplitude is around 0.25 m in $y_M$-direction for higher wind and 0.3 m for lower wind towards the tip of the blades. Both of these measures are significantly higher than the measurement uncertainty caused by the laser scanner, which has a maximum of 0.7 mm towards the tip of the blade. The measured differences are therefore actual differences and not caused by the measurement uncertainty, and the some holds for the different blade deformation according to the two wind load categories.

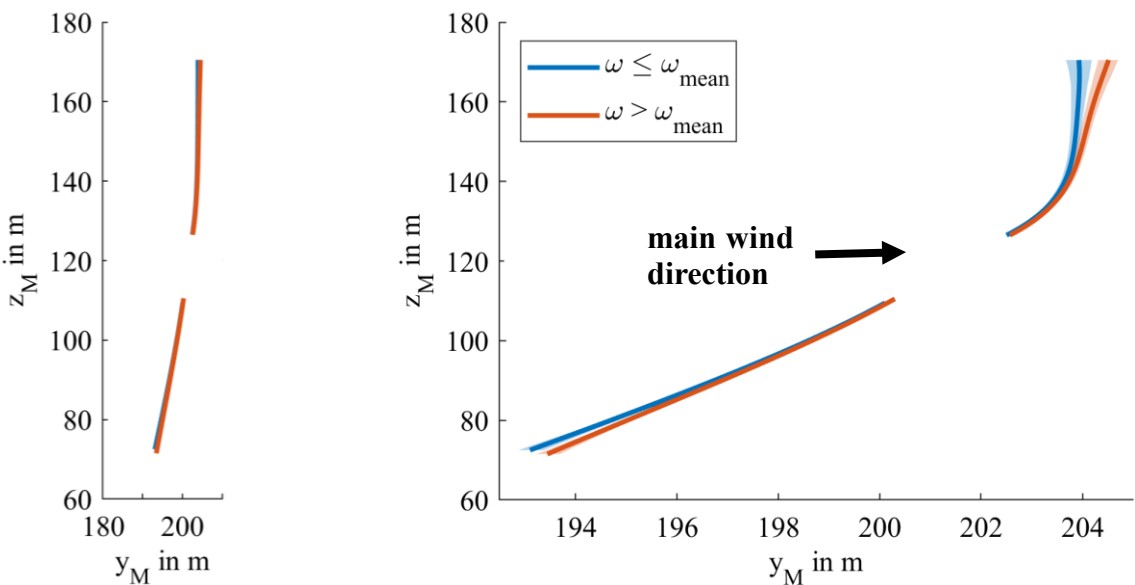

**Figure 13: Blade deformation for different wind loads for blade 1: 4ᵗʰ-degree polynomial fitted through all measurement points on the blades for higher wind load ($\omega > \omega_{\text{mean}}$) in orange and lower wind loads ($\omega \leq \omega_{\text{mean}}$) in blue, respectively, with equal spacing between the $x_M$- and $z_M$-axis (left) and distorted view (right).**

There is a visible distinction between the two wind load categories. The difference between the two deformed states increases towards the tip of the blade. Higher rotor speed is caused by a higher wind load, which pushes the blade in the wind direction

resulting in a higher deformation. Furthermore, the centrifugal forces on the rotor blades also increase with the rotor speed. In turbines with a cone angle such as the studied one, the centrifugal forces cause bending moments, which counteract the cone angle, adding to the deformation in the direction of the wind. The upper blade shows a slightly increased difference between the two cases compared to the lower blade, which can be explained by the tower behind the blades which reduces the wind load on the blade. Also, the wind load is typically lower closer to the ground. The amplitude of the oscillations decreases

slightly with added wind force, which could also be explained by the higher centrifugal force and the stiffening of the rotor blade.





The presented experiments were performed in partial load operation mode of the wind turbine to ensure the correlation between wind load and rotor speed. In order to measure the deformation during full operation mode, an additional system would be required to measure the wind speeds. Additionally, to ensure that the measured deflection of the blade results exclusively from

the wind load and is not a result of an increased measurement distance caused by a yaw of the turbine, a measurement period without yawing was chosen. The current measurement period of around 30 minutes could be reduced significantly if the blades were not analyzed separately, since there was no significant difference visible between the blades as seen in Figure 11. Nevertheless, the presented measurement method with the laser-scanner system is proven to be capable to show the different deformation behaviors of the rotor blades under varying wind loads during partial load operation.

**5.2.    Tower-blade tip clearance**

The tower-blade tip clearance is determined as described in section 2.2.3 and, as a validation, a video measurement system also measuring tower-blade tip clearance from the side is used for the second half of the measurement period (~1100 s). In Figure 13, both the tower-blade tip clearances calculated from the laser-scanner system and the video-system are shown over the measurement period in seconds. The laser-based tower-blade tip clearance measurement shows good agreement with the

video-based reference measurement, deviating only 0.06 m or 0.06 % on average with a standard deviation of 0.175 m or 1.7 % from the mean tower-blade tip clearance. The laser scanner is therefore an efficient measurement system to evaluate the tower-blade tip clearance. The differences between the minimum and the maximum measured clearance is around 1.5 m which is significantly higher than the simulated uncertainty of 0.01 m for the laser scanner. The deviation between the laser scanner and the video system can be at least partly explained by the uncertainty of the video measurement system.

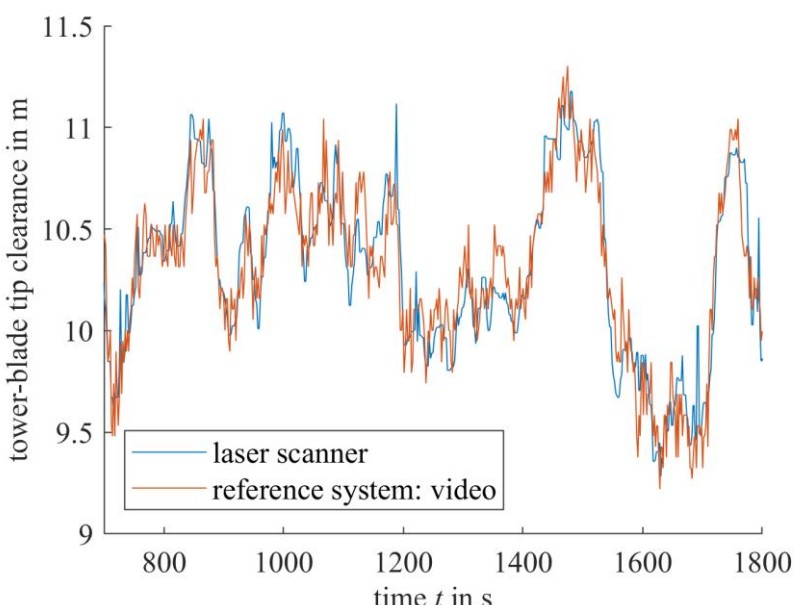


**Figure 14: Tower-blade tip clearance over time with laser scanner and video reference measurement system.**





In addition to the tower-blade tip clearance, the rotor speed $\omega$ is calculated at the same time. Figure 15 shows the measured tower-blade tip clearance from the laser scanner for the different blades (blade 1 in blue, blade 2 in orange, blade 3 in green) over the associated calculated rotor speed $\omega$ for the whole measurement period (~1800 s). Since the rotor speed can be

considered as a measure of the wind speed (part load operation), the diagram shows the relation between tower-blade tip clearance and wind load.

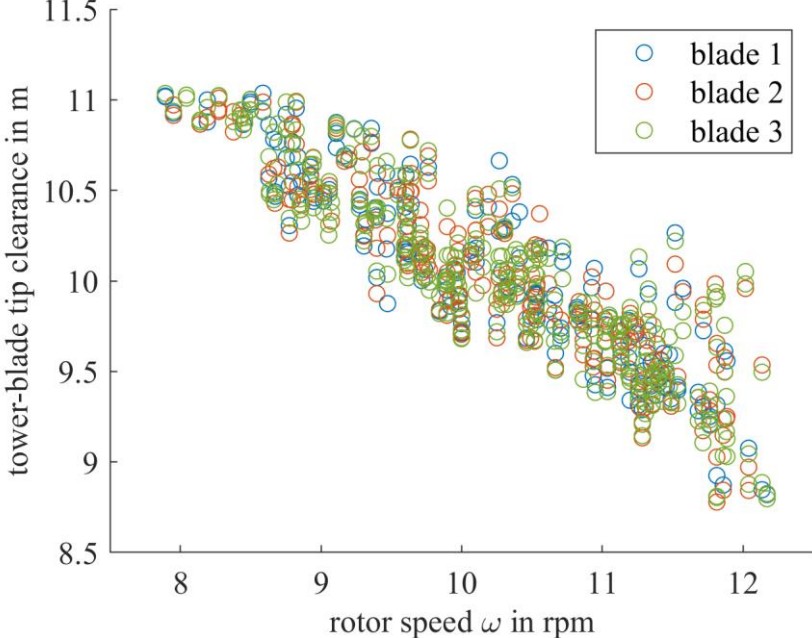

**Figure 15: Tower-blade tip clearance over rotor speed $\omega$ with laser scanner system for different blades (blade 1 in blue, blade 2 in orange and blade 3 in green)**

While no difference is visible in the behavior between the three blades, a relatively high deviation occurs, which could be caused by 10-second averaging while calculating the rotor speed. However, there is a clear correlation between the measured tower-blade tip clearance and the calculated rotor speed: The clearance decreases with rising rotor speed as the rising wind force pushes the blade towards the tower and thus decreases the distance between the two.

The comparison between the rotor speed and the tower-blade tip clearance is reasonable during partial wind load operation of

the wind turbine as is true in this case, since there the rotor speed increases linearly with the wind speed. To further study the correlation between wind loads and tower-blade tip clearance for higher wind loads, an additional wind measurement system would be necessary. Even though the uncertainty is increased compared to the rotor blade deformation analysis since only one blade scan and one tower scan was used for fitting, it is still possible to prove a clear correlation between wind loads and tower blade-tip clearance as well as measuring the tower-blade tip clearance at any given time, proving that the proposed system to

be a powerful tool in the structural health monitoring of wind turbines.



## 6. Conclusions

The proposed laser scanner-based measurement system is capable of comparing the deformed rotor blades for different wind load scenarios, as well as measuring the tower-blade tip clearance as a function of the rotor speed. This was achieved by placing the laser scanner in vertical alignment at a measuring distance of over 150 m from the wind turbine from a single
access point. Using the rotor speed as a proxy for the wind load and with the help of a least-squares fitting through the measured rotor blade data points and the measured tower points, it was possible to determine the average deformed rotor blade for different wind loads, as well as the tower-blade tip clearance. The rotor speed was calculated using only the laser scanner data, eliminating the need of an additional measurement device.

The measurement approaches were tested on an operating tall onshore wind turbine in partial load operation mode. As a result,
no behavioral difference between the blades were detected. However, there was a clear correlation between the wind speed and the deformation of the blade as well as the tower-blade tip clearance. To determine the uncertainty of the measurement approach, a Monte-Carlo simulation was carried out, where it was shown that a distinction of blade deformations is possible in the mm-range. The measured differences between the different wind load scenarios were significantly higher than the estimated uncertainty caused from the laser scanner. The measured data from the laser scanner for the tower-blade tip clearance
was validated by reference measurements with a video measurement system.

Currently, the analysis between the wind loads and the deformation or tower-blade tip clearance is only possible during partial load operation mode. By using additional data such as SCADA data or external wind measurement system, however, the measurement approach can be straightforwardly be adopted to also measure in full load operation mode. The proposed laser scanning method therefore proved to be a capable tool for the structural health monitoring of wind turbine blades.

**Author Contribution**

Conceptualization: PH, MS, AF, AvF, methodology: PH, AI, AF, KW, investigation: PH, KW, formal analysis: AI, PH, software: PH, AT, writing – original draft preparation: PH, writing – review & editing: AvF, MS, AF, supervision: MS, AF

**Competing interests**

The authors declare that they have no conflict of interest.

**Acknowledgements**

This research was funded by the German Federal Ministry for Economics and Climate Action (BMWK) within the project of PreciWind, Grant Number 03EE3013A.



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
