# Peer review of "Assessing the rotor blade deformation and tower-blade tip clearance of a 3.4 MW wind turbine with terrestrial laser scanning"

_Wind Energy Science, 2022_

## Author Response (AR1)

**Response to comments from Referee 1**

The paper presents a new approach that can be used for structural health monitoring of wind turbines and it is recommended for publication. However, there are still some minor points that should be addressed in more detail.

Thank you very much for your kind and thoughtful comments. Here are our responses to the specific comments

**Line 38:**

"(strain gauges or accelerometers) ...can lead to altered structural behavior which can impair the validity of the measurement results". This sentence is a bit misleading. Structural response may change if mass or stiffness properties of the structural components do change. Accelerometers or strain gauges have negligible mass and no contribution on the stiffness. If you perform some tests on small-size models in the lab, this statement can be correct but it is not valid for full-scale large wind turbines. If mounted on the exterior surface of the blade, maybe they can affect airflow and aerodynamic properties but these affects would also be small.

Thank you for your comment. We agree that the sentence you mentioned is misleading and we have decided to delete it from the manuscript.

**Line 71:**

Please change lane scanning to line scanning

**The wording has been corrected.**

**Line 90: Figure 1:**

Is it possible to change the scan angle or does it have to be 90 degrees? Please explain, because in Figure 1 most of the measured data seems to be useless because the laser aims at the ground not the turbine. If you can limit the angle to a smaller value (e.g. 60 degrees), maybe you can increase scanning frequency.

We agree that it would be helpful to limit the scan angle to increase the number of points that actually hit the wind turbine. Unfortunately, with the current laser scanner in use, it is neither possible to change the scan angle nor the scan frequency. We have added the idea of using a different scanner with a variable scan angle in the outlook of the manuscript.

**Line 124:**

Tower base is considered to be a fixed support. Actually, tower response seems to be better represented by a polynomial fit. If you assume that deformation will be distributed linearly, this will overestimate tower deformation and therefore underestimate tower tip clearance. I am not sure what the degree of the improvement will be. Since you had put a lot of effort in increasing the accuracy of the estimations, I would like to mention my suggestion. You do not have to change your calculations but maybe you can consider this issue for your further analyses.

We agree that a polynomial fit best describes the whole tower. However, since we are only using a small subsection of the tower (around 6 m), we decided to use a linear fit to simplify the analysis and reduce the number of variables that might lead to additional uncertainty. We have clarified this in the manuscript.

**Line 212:**

"...92 measuring points are simulated to represent the upper blade and 83 measuring points representing the lower blade". Can you please explain the difference in number measurement points?

My guess: Angle increments are constant (0.09 degrees). For small angles constant angle steps would result in large displacement steps on the structure. Therefore, you will need less number of steps to fully scan the lower blade. For large angles however, the same angle increment will result in relatively shorter segments (displacement steps) and more measurement points will be needed to scan the upper blade.

Exactly as you have pointed out in your comment, the difference in the number of points between the upper blade and lower blade is caused by the fixed angle increment since the upper blade covers a larger angle from the point of view of the laser scanner. Since we cannot change the angle increment in the laser scanner currently in use, there is no way to change this.

**Line 305**

How did you synchronize the camera and the laser measurements?

The camera and laser scanner were synchronized manually using the displayed time on both devices. A note has been added to the manuscript to clarify this.

**Line 315 Figure 14:**

How could you obtain continuous tip clearance measurement by using laser? Most probably, the graphs are step-wise but you have to mention that explicitly because in the text you have already said that clearance measurements are discontinuous. If the rotational frequency is 10 rpm (0.1667 Hz), period should be approximately 6 seconds and a blade should pass in front of the tower at every 2 seconds. So these graphs should consist of discrete points with a time interval of 2 seconds. You do not change the graph but please explain that the graph has actually a step-wise structure with a time increment of 2 seconds or more.

As you have pointed out, the graph should be displayed step-wise but was shown as continuous in order to improve visualization. We have added a line to the manuscript to clarify that the graph consists of discrete points with a time interval between 1 to 2 seconds.

**Response to comments from Referee 2**

A good manuscript.

Just a few observations:

2.2.2 The authors state: since the rotor speed  $\tilde{0}^{\bullet}$  C increases linearly with the wind speed in the turbine part-load operation mode, the rotor speed can serve as an indirect measure of the wind load.

Table 4 shows the rotor speed range and average wind speed. If the authors' hypothesis is correct, this means that:

a wind speed of 3.9 m/s corresponds to a rotor speed of 8.8 rpm (7.4 + 10.2)/2 a wind speed of 5.9 m/s corresponds to a rotor speed of 9.2 rpm (8.4 + 10.0)/2 the correlation is then: wind speed = 5 rotor speed - 40.1

Please, demonstrate a linear correlation or correct your statement.

**Thank you for your comment.**

Since this turbine is a pitch-controlled wind turbine, there is a linear (affine) correlation between wind speed and rotor speed in partial load as shown for example in (Hau, 2016, p. 488). This means that the linear correlation between the wind speed and rotor speed is only valid between the cut-in wind speed (3.5 m/s which corresponds to around 7.5 rpm) and around the rated wind speed (13.5 m/s or approximately 14 rpm). We have clarified the validity range of the linear correlation in the manuscript.

However, we decided against explicitly quantifying this linear affine relationship in the manuscript, because the focus of this paper is not this linear correlation but that the wind load increases with higher rotor speed. In addition, we only have limited data points available, and the wind speed mentioned in the manuscript in Table 4 is a rough estimate of the average wind speed during the duration of the experiment with an unknown uncertainty since we do not have access to an external wind lidar measurement system or detailed SCADA-data.

Hau, E.: Windkraftanlagen: Grundlagen. Technik. Einsatz. Wirtschaftlichkeit, 6. Aufl. 2016, Springer Berlin Heidelberg, Berlin, Heidelberg, 997 pp., 2017.